# Japanese and Bohemian Knotweeds as Sustainable Sources of Carotenoids

**DOI:** 10.3390/plants8100384

**Published:** 2019-09-28

**Authors:** Valentina Metličar, Irena Vovk, Alen Albreht

**Affiliations:** 1Department of Food Chemistry, National Institute of Chemistry, Hajdrihova 19, SI-1001 Ljubljana, Slovenia; 2Faculty of Chemistry and Chemical Technology, University of Ljubljana, Večna pot 113, SI-1000 Ljubljana, Slovenia

**Keywords:** *Fallopia japonica* Houtt, *Fallopia x bohemica*, mechanical control, renewable carotenoid source, carotenoid determination, chromatography, mass spectrometry, qualitative, quantitative analysis

## Abstract

Japanese knotweed (*Fallopia japonica* Houtt.) and Bohemian knotweed (*Fallopia x bohemica*) are invasive alien plant species, causing great global ecological and economic damage. Mechanical excavation of plant material represents an effective containment method, but it is not economically and environmentally sustainable as it produces an excessive amount of waste. Thus, practical uses of these plants are actively being sought. In this study, we explored the carotenoid profiles and carotenoid content of mature (green) and senescing leaves of both knotweeds. Both plants showed similar pigment profiles. By means of high performance thin-layer chromatography with densitometry and high performance liquid chromatography coupled to photodiode array and mass spectrometric detector, 11 carotenoids (and their derivatives) and 4 chlorophylls were identified in green leaves, whereas 16 distinct carotenoids (free carotenoids and xanthophyll esters) were found in senescing leaves. Total carotenoid content in green leaves of Japanese knotweed and Bohemian knotweed (378 and 260 mg of lutein equivalent (LE)/100 g dry weight (DW), respectively) was comparable to that of spinach (384 mg LE/100 g DW), a well-known rich source of carotenoids. A much lower total carotenoid content was found for senescing leaves of Japanese and Bohemian knotweed (67 and 70 mg LE/100 g DW, respectively). Thus, green leaves of both studied knotweeds represent a rich and sustainable natural source of bioactive carotenoids. Exploitation of these invaders for the production of high value-added products should consequently promote their mechanical control.

## 1. Introduction

Japanese (*Fallopia japonica* Houtt.) and Bohemian (*Fallopia x bohemica)* knotweeds are large herbaceous plants which belong to the Polygonaceae family [1]. In the 19^th^ century, Japanese knotweed was brought from Asia to Europe primarily for ornamental purposes. On the other hand, Bohemian knotweed emerged thereafter as a hybrid between Japanese knotweed and Giant (*Fallopia sachalinensis*) knotweed—another member of *Fallopia* spp [1,2]. Due to their fast spread and strong resilience to extermination, knotweeds were soon classified as aggressive alien plant invaders [1,2]. Today, they pose a great economic and environmental threat as they cause loss of native biodiversity, affect agriculture and forestry, and they also endanger certain animal species [2,3]. Invasive knotweeds thrive alongside water bodies and their banks, and can raise problems with water quality, accessibility, and flow rate [3,4]. These plants are also found in urban areas, on roads and railways, where they cause significant structural damage to pavements, buildings, and traffic infrastructure [4]. Bohemian knotweed is known to be more vigorous and persistent than its parents and is one of the most invasive plants in Europe [4]. 

A variety of eradication techniques exist (mechanical, chemical, and biological), out of which, mechanical excavation and incineration of rhizomes is most commonly used [5,6]. As effective as this containment method may be, a large production of waste and a high-energy demand make this means of control environmentally and economically untenable. Chemical control employs different herbicides (for example, glyphosate, imazapyr, and synthetic auxins) [6,7,8], where most of them are harmful to humans and to the environment, inefficient for large-scale application, and prohibitively expensive [6,7]. Mites, nematodes, or fungi are used to control weed through biological means. For instance, psyllid (*Aphalara itadori*) species from Japan can defoliate knotweed species substantially [9], but they can also adapt and grow on other, non-target, plants, making this approach inadequate and environmentally unsafe [5,7,8,9]. Therefore, current containment techniques have many deficiencies, so new approaches to control and suppress the invasion of knotweeds are being sought [5,6].

There is, however, also a beneficial side to knotweeds. Their flower is an important source of nectar for honeybees [10], they accumulate heavy metals from contaminated soil in their leaves and stems, and their huge biomass can be used as raw material within the paper production industry [10,11]. Knotweeds have been used in traditional Chinese medicine for decades as laxatives and food. Moreover, many pharmacological properties have been attributed to knotweeds which are used to treat inflammation, hyperlipidemia, infection, and cancer, as well as other ailments (sore throat, toothache, ulcer, hemorrhoids) [11,12].

Different plant parts of Japanese and Bohemian knotweeds are reported to contain many biologically active components such as stilbenes [11,13], anthraquinones (emodin, physcion, citreorosein, rhein, fallacinol, questin) [11,13], phenylpropanoides (lapathosides), polyphenols (flavonoids) [11], tannins [11,12], phenylpropanoide alcohols [14], and essential oils [11,12]. Sterols [12] and proanthocyanidins [13,15] are additionally reported to be present in Japanese knotweed. Carotenoids represent another important class of secondary plant metabolites [16], which play many crucial roles in living organisms [17]. For instance, oxygenated carotenoids (xanthophylls) regulate energy dissipation during photosynthesis in plants [18]. Moreover, lutein and zeaxanthin derivatives are present in the human eye macula and are responsible for maintaining good vision [19]. β-Carotene, β-cryptoxanthin, and α-carotene are pro-vitamins A [20,21], and carotenoids in general have antioxidant [20,22], anticancer [21,23], anti-inflammatory, antidiabetic, neuroprotective, and anti-obesity effect [22]. They protect skin, tissue, and cells from environmental toxins and diseases [21], they efficiently prevent cardiovascular and related diseases [21,23], and they have a beneficial effect on adipocyte differentiation [21,24]. Carotenoids are endogenous only to plants and microbes. Humans and animals are incapable of synthesizing these pigments and must obtain them through diet [20]. Total syntheses of some carotenoids are known, but they are currently not financially sustainable. Thus, there is an ongoing search for new and rich natural sources of carotenoids [25].

The main aim of this study was to evaluate Japanese and Bohemian knotweeds as sustainable sources of carotenoids. For this reason, a detailed qualitative as well as quantitative analysis of carotenoids in mature (green) and senescing knotweed leaves (yellow and green–yellowish) was carried out by means of high performance liquid chromatography (HPLC) coupled to photodiode array (PDA) and mass spectrometric (MS) detector, high performance thin-layer chromatography (HPTLC) with densitometry and image analysis, and gas chromatography (GC) coupled to MS.

## 2. Results and Discussion 

### 2.1. Identification of Carotenoids in Knotweed Leaf Extracts

Japanese and Bohemian knotweed leaf extracts were screened for carotenoids by HPTLC (Figure 1).

Japanese and Bohemian knotweed green leaf extracts and spinach leaf extract showed similar chromatographic profiles, but unlike the latter, knotweeds contained some additional yellow-colored pigments in the R_F_ region 0.84–1 (Figure 1A). These were tentatively identified as phenolic compounds by comparison of HPTLC–MS^2^ and HPTLC–MS^3^ spectra with literature data [26]. Main yellow pigments of knotweeds were identified by standard co-migration as (all-*trans*)-β-carotene (R_F_ = 0.16) and (all-*trans*)-lutein and/or (all-*trans*)-zeaxanthin (R_F_ = 0.55). As expected, green bands representing chlorophyll a/a’ and chlorophyll b/b’ (Figure 1A, tracks 2, 4, and 5; R_F_ region 0.30–0.45) were absent in both yellow (senescing) Japanese and Bohemian knotweed leaf extracts. However, additional yellow bands appeared in the non-polar R_F_ region 0.02–0.15 (Figure 1A, tracks 1 and 3). These were assumed to be carotenes or xanthophyll esters. A chemical profile transition between green- and yellow-colored plant leaves was evident from the chromatogram of green–yellowish leaf extract, which contained compounds of green as well as of yellow leaf extracts (Figure 1A, track 6).

To further identify carotenoid constituents of knotweeds, the developed HPTLC plate was exposed to HCl vapor (Figure 1B). A change in band color from yellow or orange to blue or green in the presence of a strong acid is generally indicative of an epoxide functional group within the carotenoid structure. A bathochromic shift in the absorption spectrum is a result of the formation of an oxonium ion at the terminal end of the carotenoid-conjugated double bond chain [27,28]. Thus, diepoxides give deep blue color and monoepoxides give a green–blue color [29]. In chromatograms of green leaf extracts of both knotweeds, the blue coloration of TLC bands indicated two epoxide carotenoids at R_F_ = 0.65 and 0.73 (Figure 1B, tracks 2, 4, and 5). In senescing leaves, highly hydrophobic carotenoids with R_F_ < 0.15 were identified as epoxides as well. A drop in absorbance for (all-*trans*)-lutein/zeaxanthin (R_F_ = 0.55) and (all-*trans*)-β-carotene (R_F_ = 0.16) was observed after HCl exposure, but there were no significant changes in the measured absorption spectra in the range 190–800 nm, despite an appreciable darkening of those bands (Figure 1B, tracks 2, 4, and 5). Under strongly acidic conditions such as those used here, carotenoids are prone to isomerization and degradation, or may even react with co-migrating compounds and plate impurities, which could explain the reduced absorption and changes in band color. HPTLC–MS analysis of bands immediately after plate development was also attempted, but co-migration of compounds and stationary phase additives produced a high MS background, which, for the most part, impeded the carotenoid structural elucidation and quantitation. 

In order to gain the extra separation resolution, knotweed leaf extracts were analyzed by HPLC–PDA–MS. A ProntoSIL C30 column was used because it enables the differentiation even between all-*trans* and *cis* isomers of pigments [30,31]. Analytes were identified by comparing the absorption and MS^n^ spectra with the literature data, by co-elution with standards, and by relative hydrophobicity of eluted compounds. Again, chromatographic profiles of spinach and knotweed green leaf extracts were very similar and less complex in comparison to yellow leaf extracts (Figure 2, Table 1). all-*trans*-Lutein, all-*trans*-β-carotene, neoxanthin, and violaxanthin were identified as major carotenoid constituents of green leaf extracts of Japanese and Bohemian knotweeds, representing roughly 86% and 82%, respectively, of the total carotenoid content (Table 2). Luteoxanthin, all-*trans*-zeaxanthin, antheraxanthin, and (9-*cis*)- and (13-*cis*)-β-carotene were found at lower levels. Luteoxanthin was differentiated from its structural isomers violaxanthin and neoxanthin by differences in absorption spectra; with a shorter conjugated chain of double bonds in its structure, the absorption maximum of luteoxanthin was blue-shifted by 20–25 nm compared to the other two epoxycarotenoids. Geometric isomers of β-carotene were confirmed by a controlled isomerization of the all-*trans* form with iodine [32]. Moreover, both (9-*cis*)- and (13-*cis*)-β-carotene showed a small but characteristic hypsochromic spectral shift of a few nanometers and the appearance of the *cis* absorption peak in the 330–338 nm region. The green–yellowish leaf extract of Japanese knotweed showed a transition between the “all-green” and “all-yellow” leaf chromatographic profiles (Figure 2E). The concentration of free carotenoids diminished (most profoundly for the epoxide violaxanthin), while carotenoid esters appeared. In yellow senescing leaves, free carotenoid content was even lower and amounted to less than 67% and 42% of the total pigments for Bohemian and Japanese knotweed, respectively. The remainder of pigments was in the form of esters (Table 1), which contained an epoxide carotenoid at the structural core (mainly violaxanthin). There was one exception, tentatively identified either as zeinoxanthin palmitate oleate or β-cryptoxanthin palmitate oleate. The presence of epoxide esters was in line with the blue coloration of bands seen in the hydrophobic region of the HPTLC plate (0.02 < R_F_ < 0.15) after HCl exposure (Figure 1B). Esterification of xanthophylls is known to coincide with chloroplast degradation in autumn and it is a plant’s way of adapting to the changes in its metabolic system. Enhancement of xanthophyll’s lipophilicity supposedly increases their stability and solubility in lipid-rich plastoglobules once the thylakoid membrane gets disrupted. Epoxides react more rapidly with fatty acids [27,33], so this might explain why mainly esters of violaxanthin and antheraxanthin were found in senescing leaves of both knotweeds. For Bohemian knotweed, the ester-forming fatty acids were determined as palmitic, myristic, stearic, oleic, and lauric acids. GC–MS analysis additionally confirmed their identity by co-elution with standard compounds and/or by a comparison of the obtained MS data with the NIST library. Given the high correlation of chromatographic profiles of both knotweeds (Figure 2D,F), it is safe to assume that these fatty acids also form esters with carotenoids from Japanese knotweed.

Apart from being present in green leaf extracts, (9-*cis*)-β-carotene was also found in trace amounts in senescing leaf extracts. Carotenoids occur naturally in the all-*trans* geometric form, but they are prone to isomerization. Upon heating and light exposure, all-*trans*-β-carotene is particularly unstable and readily converts into one of its *cis*-isomers. At equilibrium, their relative abundances are in the following order: all-*trans* > 9-*cis* > 13-*cis* > 15-*cis* [34]. Low amounts of violaxanthin and luteoxanthin *cis* isomers were also determined (Table 1). α-Carotene and free β-cryptoxanthin were not detected in any of the studied leaf extracts. Finally, chlorophylls, visualized as green bands on the HPTLC plate, were baseline resolved by means of HPLC and identified as chlorophyll a/a’ and b/b’. Interestingly, TLC–MS^2^ and MS^3^ analyses of green bands revealed only pheophytin a and b (chlorophyll lacking Mg^2+^), most probably due to the intrinsic acidity and polarity of the silica-based adsorbent. In this study, green and senescing leaves of Bohemian and Japanese knotweeds were comprehensively explored for their carotenoid content for the first time. A tentative identification and quantitation of carotenoids from green leaves of Japanese knotweed was attempted before by Lachowicz et al. [35], but the number and identity of detected pigments differ significantly between the studies. A C18 stationary phase used by Lachowicz et al. is often ineffective of separating certain isomeric pigments, which might have led to their co-elution and detection failure. Moreover, the addition of an acidic mobile phase modifier (formic acid in their case) presumably led to low carotenoid recovery yields and their on-column transformations, e.g., the conversion of neoxanthin into neochrome [36].

### 2.2. Quantitation of Carotenoids

Carotenoids are prone to degradation and isomerization when exposed to oxidants, high temperature, light, metals, and acids. Therefore, special attention should be given to experimental conditions during all stages of qualitative and quantitative analysis. Synthetic antioxidants are often employed to enhance the stability of pigments [20]. Interestingly, addition of 2-tert-butylhydroquinone (TBHQ) (0.1% *w/v*) to the extraction solvent did not increase the extraction recovery of carotenoids in our case. Triethylammonium acetate (TEAA) was also used before as an extraction solvent additive and as a mobile phase modifier in HPLC [42]. In our case, the presence of 15% 1M TEAA slightly increased carotenoid recovery (4 ± 1%; *n* = 3). Moreover, it was previously reported that, in terms of recovery, ultrasound-assisted solid–liquid extraction of carotenoids is inferior to solid–liquid extraction where the extraction is performed by means of mere stirring [37]. However, we observed that when reduced light and the absence of oxygen was ensured during extraction, both procedures delivered comparable results (<1% difference; RSD = 1.2%, *n* = 3). This demonstrated that short-term exposure of carotenoids to high temperatures (several thousand °C presumed at local hot spots created during sonication) did not seem to have a detrimental effect on pigment degradation and, on the other hand, there was no evident increase in extraction recovery resulting from a more efficient disruption of chloroplast membranes by ultrasonic waves. Both contributions could eventually balance one another, resulting in the observed zero net effect on recovery. Further work is needed to pinpoint the major individual factors that govern extraction recovery, but this was outside of the scope of this study. Since solid–liquid extraction by stirring enabled better temperature control and a higher throughput in our case, and since ultrasonic assisted extraction had no clear advantages, the latter was not pursued any further.

Recoveries of carotenes and xanthophylls from plant materials by using 90% acetone (aqueous):1 M TEAA (85:15, *v/v*) were approximated as recoveries of (all-*trans*)-β-carotene and (all-*trans*)-lutein, respectively, and are presented in Table 2.

Good extraction recoveries of (all-*trans*)-lutein were obtained for all studied leaf extracts (>81%). Recovery of (all-*trans*)-β-carotene from green leaf extracts was also excellent (>85%); however, for senescing leaf extracts, these values were as low as 54%. These low recoveries are presumably a reflection of the changes in the chemical profile of senescing leaves. We assume that higher levels of endogenous acids, which ironically increase stability of xanthophylls through natural esterification, should have an adverse effect on the stability of free carotenoids. This is especially true for a more labile (all-*trans*)-β-carotene. 

Total concentration of carotenoids in green leaf extracts of both knotweeds were appreciably higher compared to the senescing leaf extracts (Table 2). The results show that with chloroplast decomposition, degradation of not only green chlorophylls, but also carotenoids occurs. However, among non-esterified carotenoids, (all-*trans*)-zeaxanthin is a clear exception with an increase in its concentration in autumn leaves. These values, relative to its isomer (all-*trans*)-lutein, were still low, but significant, which indicates an alternative biological role of this pigment in autumn. Qualitatively and quantitatively, there were only minor differences in carotenoid profiles between the two studied knotweed species. Total contents of carotenoids in green leaf extracts of Japanese knotweed (378 mg/100 g DW) and Bohemian knotweed (260 mg/100 g DW) were comparable to the total carotenoid content determined in spinach (384 mg/100 g DW)—a well-established rich source of these biologically active secondary metabolites.

The average total carotenoid content of green leaves of Japanese knotweed measured here (Table 3) was approximately three times higher compared to a study on Japanese and Giant knotweed reported by Lachowicz et al. [35]. The reason for this discrepancy most probably lies in the different sample preparation and analytical procedures; here, controlled experimental conditions (reduced light, inert atmosphere) were used to ensure the stability of labile analytes [36]. Compared to common food sources rich in (all-*trans*)-lutein and (all-*trans*)-β-carotene (Table 2), green leaves of Japanese and Bohemian knotweeds present an excellent potential alternative. Marigold flower (*Tagetes erecta* L.), which is being massively cultivated on plantations for the production of lutein-containing food supplements, contains comparable amounts of (all-*trans*)-β-carotene and has only approximately 2-fold higher levels of (all-*trans*)-lutein.

## 3. Materials and Methods 

### 3.1. Chemicals and Standards

All solvents and chemicals were at least of analytical grade. Acetone (HPLC grade), methanol, 2-tert-butylhydroquinone (TBHQ, 97%), and 1 M aqueous solution of TEAA buffer (pH = 7) were supplied by Honeywell (Seelze, Germany). Dichloromethane, acetic acid (100%), *n*-hexane, and acetone (LC–MS grade) were purchased from Merck (Darmstadt, Germany). 2-Tert-butylhydroquinone (TBHQ, 97%) was from Sigma-Aldrich (St. Louis, MO, USA). Boron trifluoride solution (BF_3_, 1.3 M) in methanol was from Supelco Analytical (Bellefonte, PA, USA). Lutein (≥95%), zeaxanthin (≥98%), β-carotene (≥98%), and β-cryptoxanthin (≥97%) were purchased from Extrasynthèse (Genay, France). α-carotene (≥98%) was from Sigma-Aldrich. Stearic (≥99.5%), myristic (≥99.5%), linoleic (99%), capric (≥99.5%), palmitic (≥99%), and oleic (99%) acids were purchased from Fluka Chemika (Steinheim, Germany). MilliQ water (18.2 MΩ) was used throughout. 

### 3.2. Preparation of Carotenoid Standard Solutions

Carotenoid standards (1 mg) were individually weighed into 50 mL flasks. Dichloromethane (2 mL) was added to facilitate dissolution and then the flask was filled up to the volume mark with acetone which was previously deaerated by sparging with nitrogen for 30 min. Working standard solutions were prepared by further dilution of each standard stock solution (0.02 mg/mL) with acetone. Exact concentrations of working standards were determined spectrophotometrically using the following relationship:c = A_(λmax)/_A_(1%,1cm,λ)_(1)
where c is concentration in μg/mL, A_(λmax)_ is the measured absorbance, and A_(1%,1cm,λ)_ is the specific absorption coefficient for the selected analyte in acetone, taken from the CaroteNature certificate (2540 for lutein, 2500 for β-carotene, and 2350 for zeaxanthin) [52,53]. All standard solutions prepared in this manner were stored in amber glass vials (National Scientific Company, USA) at –80 °C prior to use.

### 3.3. Plant Materials and Preparation of Leaf Extract Solutions for the Analysis of Carotenoids and Fatty Acids

Fresh yellow, green, and green–yellowish colored leaves of Japanese and Bohemian knotweeds harvested in the area of Ljubljana, and fresh green leaves of spinach purchased at a local market, were frozen in liquid nitrogen and lyophilized by Micro Modulyo (IMA Edwards, Bologna, Italy) for 24 h at –50 °C. Dry material was afterwards again frozen in liquid nitrogen and pulverized by Mikro-Dismembrator S (Sartorius, Göttingen, Germany) at 1700 min^−1^ for 1 min. Pulverized material was transferred into amber glass vials and stored at –80 °C prior to use. 

Extraction of carotenoids was carried out by accurately weighing the pulverized plant material (20–30 mg) into 45 mL glass tubes. Then, 10 mL of 90% acetone (aqueous):1 M TEAA (85:15, *v/v*) was added and the mixture was stirred in the Carousel 12 Plus apparatus (Radleys, Safron Walden, UK) for 15 min at room temperature under nitrogen atmosphere and reduced light conditions. Afterwards, leaf extracts were centrifuged at 1800 g for 5 min. Supernatants were filtered through a 0.45-μm polyvinylidene fluoride (PVDF) (LLG labware, Meckenheim, Germany) membrane and stored at –80 °C prior to use. All leaf extract solutions were prepared in triplicate.

To determine the extraction recovery of carotenoids from pulverized leaves, samples were spiked with known amounts of (all-*trans*)-β-carotene (60 mg/L) and (all-*trans*)-lutein (112 mg/L) in acetone prior to sample preparation. Standard additions were made at 100% level and the details are summarized in Table 4.

Controls were prepared by adding the same volume of acetone instead of (all-*trans*)-β-carotene and (all-*trans*)-lutein standard solution to the pulverized leaves. The spiked and control leaf solutions were then further processed by the extraction procedure described above and stored at –80 °C prior to analysis. All leaf extract solutions were prepared in duplicate. The recovery was calculated according to the following relationship:Recovery (%) = (A_sp_ – A_c_)/(A_STD_ x 100)(2)
where A_sp_, A_c_, and A_STD_ denote analyte peak areas in the chromatogram of a spiked and control leaf extracts, and in the chromatogram of a standard solution at the concentration level of the spike, respectively.

For qualitative GC analysis of fatty acids, pulverized yellow leaves of Bohemian knotweed (900 mg) were weighed into a glass beaker, *n*-hexane (40 mL) was added, and the suspension was stirred for 1 h. After the extraction, the solution was filtered through a black ribbon filter paper (pore size 12–15 μm, from Sartorius, Germany) and the solvent was removed under reduced pressure. Afterwards, *n‑*hexane (1 mL), BF_3_ in methanol (1 mL; 1.3 M), and Na_2_SO_4_ (10 mg) were added to the solid residue (or individual fatty acid standard in the case of standard preparation (5–10 mg)). Leaf extracts and standard solutions were transferred to 4-mL amber vials, capped, vortexed for 1 min, and then heated at 90 °C for 1 h to carry out the transesterification. Afterwards, the upper *n*-hexane layer was transferred to a GC vial and submitted to GC–MS analysis.

### 3.4. HPTLC with Densitometry Analysis

HPTLC analyses were performed on 20 × 10 cm glass‑packed C18 RP HPTLC silica gel plates with 0.20 mm layer thickness (Merck, Art. No. 1.05914.0001). Prior to use, plates were pre-developed with methanol:dichloromethane (1:1, *v/v*) and then dried in the oven for 20 min at 100 °C [37]. Leaf extracts and standard solutions were applied on the plates as 8 mm bands, 15 mm from the side edge, 10 mm from the bottom edge, and 10 mm apart by means of Linomat 5 or Automatic TLC Sampler 4 (samples: 50 μL—leaf extracts of spinach and green leaves of both knotweeds, 75 μL—leaf extracts of senescing leaves; standards: 200 ng of lutein, zeaxanthin, and 500 ng of β-carotene) from Camag (Muttenz, Switzerland). The plates were developed at ambient temperature in a saturated (30 min) twin trough chamber (20 x 10 cm, Camag) with developing solvent 0.1% TBHQ in methanol:acetone (1:1, *v/v*) [37]. The developing distance 7 cm was achieved in 12 min. Developed plates were dried under a stream of cool air using a hair dryer. The plates were documented by Camag Digistor 2 documentation system (Camag) under white light transmission mode and, after that, scanned at 450 nm by Camag TLC scanner 3 in absorption/reflectance mode. Spectra of bands were also recorded in the range from 190 to 800 nm. Scanning speed was 10 nm/s and slit dimensions were: Length 6 mm, width 0.4 mm. Instruments were controlled by WinCATS software (Version: 1.4.9.2001). The same plates were afterwards also exposed to HCl vapor for 20 s (epoxide test) [29] in a twin trough chamber and, after that, documented and scanned again as described above.

### 3.5. HPTLC–MS^n^ Analysis

For a direct coupling of HPTLC to an ion trap LTQ Velos MS system (Thermo Fisher, San Jose, Ca, USA), a TLC–MS interface (Camag) was used. Elution of zones of interest from the plates was performed by using methanol:dichloromethane (3:1, *v/v*) as an elution solvent. The flow rate was maintained at 200 μL/min and 0.2% acetic acid in methanol was added to the HPTLC effluent in a ratio of 1:40 prior to the introduction of the solution into the MS system. Atmospheric pressure chemical ionization (APCI) in positive ion mode was used. Ion source parameters were set as: Transfer capillary and vaporizer temperature 300 °C, sheath gas flow rate 30 a.u. (arbitrary units), auxiliary gas 10 a.u., spray voltage 3 kV, and discharge current 3 μA. MS^n^ experiments were carried out by fragmentation of target precursor ions (isolation width 2 amu) using different collision energies (30–45%). MS and MS^n^ spectra were acquired in the 200–1500 *m/z* range. The MS background was subtracted by sampling the HPTLC plate between tracks at the same R_F_ value as the measured band. Acquired MS spectra were processed using Xcalibur software (Version 2.1.).

### 3.6. HPLC–PDA Analysis

For HPLC analyses, an HPLC Surveyor Plus system from Thermo Finnigan was used, which was equipped with a thermostated autosampler (Autosampler Surveyor Plus), a quaternary pump (Surveyor LC Pump Plus), and a diode-array detector (Surveyor PDA Plus) with a 5 cm LightPipe flow cell. Separations were performed on a ProntoSIL C30 column (250 x 4.6 mm i.d., 5 μm) from Bischoff (Leonberg, Germany), connected to a security guard cartridge (C18 4 x 3 mm i.d.) from Phenomenex (Torrance, CA, USA). The mobile phase consisted of acetone (A) and 0.1 M TEAA in water (B), and the following gradient was applied: 0–12 min 90% A, 12–25 min 90–100% A, 25–30 min 90–100% A, 30–35 min 90% A. The flow rate was maintained at 0.8 mL/min, while column oven and autosampler temperatures were set at 35 °C and 15 °C, respectively. Injection volume was 10 μL. Acquisition wavelength was set to 450 nm. Spectra were also acquired in the range from 195–790 nm. ChromQuest 5.0 software was used for data evaluation. Quantitation of (all-*trans*)-lutein, (all-*trans*)-zeaxanthin, and (all-*trans*)-β-carotene was done by a five-point linear external standard calibration method using the appropriate standards. Levels of all other identified (and unknown) carotenoids were estimated by using the (all-*trans*)-lutein external standard calibration curve. For these analytes, a general relative response factor was calculated (in reference to (all-*trans*)-lutein) as a ratio between an average carotenoid absorption coefficient (A_(1%,1cm,λ)_ = 2500) [54] and the specific absorption coefficient of (all-*trans*)-lutein in acetone (A_(1%,1cm,λ)_ = 2540) [55]. In these cases, data were acquired at absorption maxima (λ_max_) of individual quantified analyte. Levels of esters were calculated by assuming the same molar absorption coefficient as their free carotenoid analogues. Total carotenoid content was expressed as milligrams of (all-*trans*)-lutein equivalent per 100 grams of dry leaf weight (mg LE/100 g DW). All quantitation analyses were carried out in triplicate.

### 3.7. HPLC–MS^n^ Analysis

The HPLC–MS system comprised an Accela 1250 UHPLC system (Thermo Fisher) coupled to an LTQ Velos MS system (Thermo Fisher) with an APCI ion source operating in positive ion mode. The HPLC system consisted of a thermostated Accela autosampler with a 25-μL loop, a quaternary high-pressure Accela pump, and a diode array Accela PDA detector. Xcalibur software (Version 2.1.) was used for evaluation of chromatograms. All HPLC conditions were the same as stated above for HPLC–PDA analysis, only solvent B was replaced with water. APCI ion source conditions were as follows: Transfer capillary temperature 275 °C, vaporizer temperature 300 °C, spray voltage 5 kV, discharge current 3 μA, sheath gas flow rate 35 a.u., and auxiliary gas flow rate 5 a.u. MS and MS^n^ spectra were acquired in a 200–1200 *m/z* range. For fragmentation of selected ions, different collision energies (30–45%) were applied.

### 3.8. GC–MS Analysis

GC–MS analyses were carried out using a GC Ultra-system (Thermo Electron Corporation) equipped with a DSQ II MS detector. Compounds were separated on a ZB-5HT column (20 m × 0.18 mm i.d. × 0.18 μm thickness) from Phenomenex. Inlet temperature was set at 290 °C and He was used as a carrier with a constant flow rate of 1 mL/min. Ion source temperature was set at 200 °C. Injection volume was 1 μL with a split ratio 1:30. Temperature gradient: Held at 150 °C for 1 min, 5 °C/min to 180 °C, 1 °C/min to 190 °C, 10 °C/min to 240 °C. Electron ionization (70 eV) ion source in positive ion mode was used and MS data was acquired in the *m/z* 50–650 range. The compounds were identified by fatty acid standard co-elution and by comparison of the acquired data with the NIST MS library (version 2.3).

## 4. Conclusions 

To conclude, carotenoids from mature and senescing leaves of Japanese and Bohemian knotweeds were explored, and we show that these plants contain a variety of carotenoids, but mainly (all-*trans*)-lutein, violaxanthin, and (all-*trans*)-β-carotene. In autumn, the leaves turn yellow due to chlorophyll degradation; however, the levels of carotenoids were shown to drop significantly as well (by more than 80%) and a major proportion of these pigments was found in the esterified form. Nonetheless, with a high total carotenoid content of 260–380 mg LE/100 g DW for mature (green) leaves, we propose Japanese and Bohemian knotweeds as new sustainable sources of carotenoids. Marigold flower (*Tagetes erecta* L.), a gold standard in the field of lutein production, contains high levels of carotenoids, but the existence of its vast plantations are now being questioned, because these could alternatively be used for cultivation of food crops for the ever growing human population. On the other hand, a large-scale harvesting of knotweeds and exploitation of their plant parts (zero waste) are encouraged and should present a powerful and economically-driven method of mechanical control of these highly invasive alien plant species.

## Figures and Tables

**Figure 1 plants-08-00384-f001:**
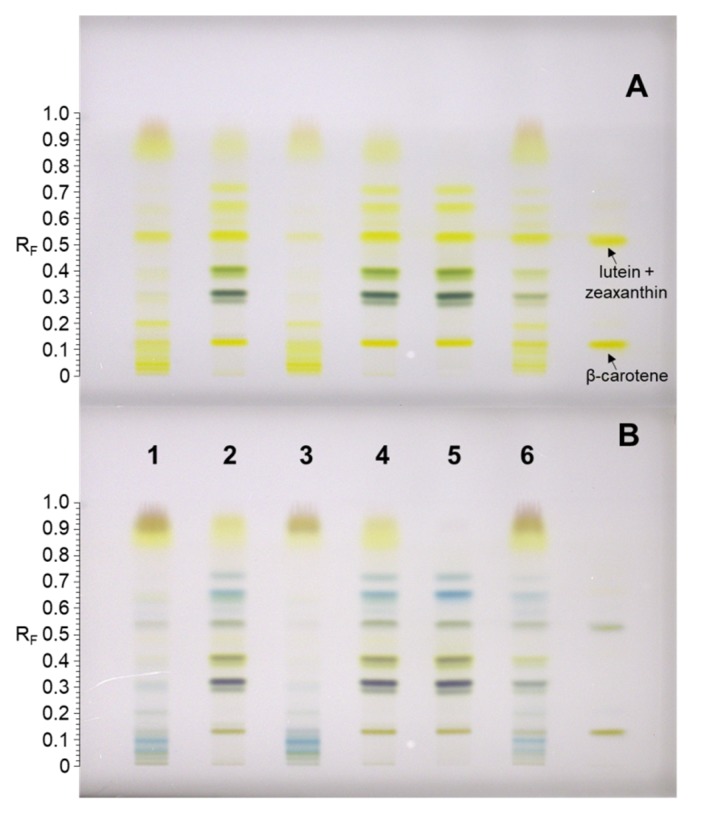
Chromatograms of leaf extracts on C18 HPTLC silica gel plates before (**A**) and after (**B**) exposure to HCl. Plates were developed with acetone:methanol (1:1 *v/v*) + 0.1% TBHQ in a saturated twin trough chamber and the images were acquired using white light in transmission mode. Plates were predeveloped with MeOH:dichloromethane 3:1 (*v/v*). Tracks: Yellow leaves of Bohemian knotweed (**1**), green leaves of Bohemian knotweed (**2**), yellow leaves of Japanese knotweed (**3**), green leaves of Japanese knotweed (**4**), spinach leaves (**5**), green–yellowish leaves of Japanese knotweed (**6**), and standard solution mix of (all-*trans*)-lutein (200 ng), (all-*trans*)-zeaxanthin (200 ng), and (all-*trans*)-β-carotene (500 ng) (**7**).

**Figure 2 plants-08-00384-f002:**
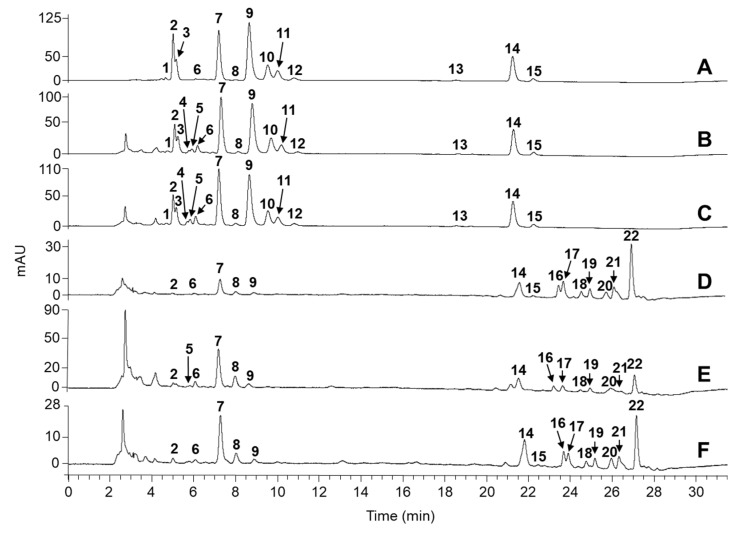
Separation of carotenoids extracted from spinach leaves (**A**), green leaves of Bohemian knotweed (**B**), green leaves of Japanese knotweed (**C**), yellow leaves of Japanese knotweed (**D**), green­yellowish leaves of Japanese knotweed (**E**), and yellow leaves of Bohemian knotweed (**F**): Violaxanthin (*cis* or *trans*) (**1**,**2**), neoxanthin (**3**), luteoxanthin (*cis* or *trans*) (**4**,**5**), antheraxanthin (**6**), (all-*trans*)-lutein (**7**), (all-*trans*)-zeaxanthin (**8**), chlorophyll b (**9**), chlorophyll a (**10**), chlorophyll b’ (**11**), chlorophyll a’ (**12**), (13-*cis*)-β-carotene (**13**), (all-*trans*)-β-carotene (**14**), (9-*cis*)-β-carotene (**15**), violaxanthin palmitate oleate (**16**), antheraxanthin dilaurate (**17**), luteoxanthin dimyristate (**18**), luteoxanthin palmitate oleate (**19**), zeinoxanthin palmitate oleate or β-cryptoxanthin palmitate oleate (**20**), violaxanthin palmitate stearate (**21**), violaxanthin myristate (**22**).

**Table 1 plants-08-00384-t001:** Constituents of leaf extracts, identified by HPLC-PDA–MS^2^ analysis.

						Japanese Knotweed			Bohemian Knotweed		Spinach	
Peak no.	t_R _(min)	Compound	Precursor Ion (*m/z*)	Fragment Ions (*m/z*)	UV/vis Absorption Maxima (nm)	Green Leaves	Yellow Leaves	Green–Yellowish Leaves	Green Leaves	Yellow Leaves	Green Leaves	Ref.
1	4.6	violaxanthin (*cis *or *trans*)	601	**583**, 565, 509, 491	475, 440, 415	+			+		+	[37,38,39]
2	5.1	violaxanthin (*cis *or *trans*)	601	**583**, 565, 517, 495	475, 442, 416	+	+	+	+	+	+	[37,38,39]
3	5.3	neoxanthin	601	**583**, 565, 413	473, 438, 412	+			+		+	[37,38,39]
4	5.5	luteoxanthin (*cis *or *trans*)	601	**583**, 565, 509, 491	450, 425, 400	+		+				[39]
5	5.8	luteoxanthin (*cis *or *trans*)	601	**583**, 565, 491	450, 425, 400	+		+	+			[39]
6	6.1	antheraxnthin	586	**567**, 554, 536, 493	470, 445, 421	+	+	+	+	+	+	[37,38,39]
7	7.2	(all-*trans*)-lutein	569	**551**, 533, 495	475, 450, 424	+	+	+	+	+	+	[37,38,39]
8	8.1	(all-*trans*)-zeaxanthin	569	**551**, 495, 449	480, 455, 435	+	+	+	+	+	+	[37,38,39]
9	8.8	chlorophyll b	907	**629**, 597, 569	645, 595, 455	+	+	+	+	+	+	[37,40,41]
10	9.7	chlorophyll a	893	**615**, 555, 538	660, 614, 430	+			+		+	[37,40,41]
11	10.0	chlorophyll b’	907	**629**, 607, 569	645, 455, 430	+			+		+	[40]
12	10.7	chlorophyll a’	893	**615**, 555, 535, 409	660, 615, 430	+			+		+	[40]
13	18.6	(13-*cis*)-β-carotene^a^	537	**481**, 413, 399	475, 450, 335	+			+		+	[41]
14	21.6 -21.8	(all-*trans*)-β-carotene^b^	537	481, **413**, 399, 347	665, 480, 455, 410	+	+	+	+	+	+	[37,38,39]
15	22.3	(9-*cis*)-β-carotene^a^	537	**481**, 413, 399, 347	475, 450, 425, 335	+	+	+	+	+	+	[41]
16	23.6	violaxanthin palmitate oleate	1103	**1085**, 543, 826, 547	470, 440, 415		+	+		+		[39]
17	23.9	antheraxanthin dilaurate	949	**932**, 547	465, 440, 415		+	+		+		[39]
18	24.7	luteoxanthin dimyristate	1006	988, **826**, 549	450, 425, 400		+	+		+		[39]
19	25.2	luteoxanthin palmitate oleate	1103	**1085**, 826, 547	450, 425, 400, 395		+	+		+		[39]
20	25.9	zeinoxanthin palmitate oleate or β-cryptoxanthin palmitate oleate	790	697, **535**	475, 450, 422		+	+		+		[39]
21	26.3	violaxanthin palmitate stearate	1100	**1081**, 844, 804, 565	465, 435, 411		+	+		+		[39]
22	27.0	violaxanthin myristate	811	794, **533**	475, 445, 423		+	+		+		[39]

^a^ Confirmed by a controlled isomerization of β-carotene with iodine [32]. ^b^ (all-*trans*)-β-carotene was confirmed by spiking the studied leaf extracts with (all-*trans*)-β-carotene standard. Slight retention time shifts of (all-*trans*)-β-carotene between leaf extracts presumably occurred due to different leaf matrices. Base fragment ions are in boldface.

**Table 2 plants-08-00384-t002:** Average extraction recovery and contents of carotenoids found in Japanese and Bohemian knotweed leaf extracts and spinach leaf extract. Limit of quantitation (LOQ) for (all-*trans*)-lutein and for analytes which were quantified based on (all-*trans*)-lutein calibration was determined as 0.8 mg/100 g (S/N > 10, RSD (%) < 1). Recovery and total carotenoid content is expressed as average of duplicate and triplicate measurements, respectively, with standard deviation.

		Japanese Knotweed			Bohemian Knotweed		Spinach
		Green Leaves	Yellow Leaves	Green–Yellowish Leaves	Green Leaves	Yellow Leaves	Green Leaves
		Recovery (%)					
**(all-*trans*)-lutein**		86 ± 2	93 ± 4	81 ± 1	93 ± 3	88 ± 5	94 ± 4
**(all-*trans*)-β-carotene**		91 ± 4	56 ± 5	72 ± 4	89 ± 1	54 ± 1	85 ± 2
Peak no.	Compound	**Content (mg/100 g DW)**					
1	violaxanthin (*cis* or *trans*)	4.9 ± 0.9	< LOQ	< LOQ	3.9 ± 0.2	< LOQ	7.1 ± 0.4
2	violaxanthin (*cis *or *trans*)	58.3 ± 7.0	< LOQ	4.2 ± 1.2	39.9 ± 0.9	1.5 ± 0.2	96.8 ± 1.8
3	neoxanthin	38.2 ± 6.5	< LOQ	3.3 ± 1.3	24.4 ± 0.6	< LOQ	44.3 ± 2.6
4	luteoxanthin (*cis *or *trans*)	2.9 ± 0.3	< LOQ	< LOQ	2.2 ± 0.1	< LOQ	< LOQ
5	luteoxanthin (*cis *or *trans*)	6.3 ± 1.1	< LOQ	1.1 ± 0.1	5.4 ± 0.4	< LOQ	< LOQ
6	antheraxnthin	10.3 ± 0.5	1.0 ± 0.1	6.4 ± 0.1	12.8 ± 0.7	2.0 ± 0.1	3.6 ± 0.1
7	(all-*trans*)-lutein	144.3 ± 8.7	9.4 ± 1.2	55.8 ± 7.7	97.1 ± 4.0	28.6 ± 2.8	127.9 ± 1.4
8	(all-*trans*)-zeaxanthin	3.4 ± 0.2	1.8 ± 0.1	5.1 ± 1.3	2.7 ± 0.1	6.1 ± 1.8	< LOQ
13	(13-*cis*)-β-carotene	1.4 ± 0.5	< LOQ	< LOQ	0.9 ± 0.2	< LOQ	1.9 ± 0.8
14	(all-*trans*)-β-carotene	97.3 ± 1.7	8.0 ± 0.5	23.2 ± 4.7	68.7 ± 0.4	12.7 ± 2.4	97.4 ± 0.7
15	(9-*cis*)-β-carotene	8.6 ± 1.7	< LOQ	< LOQ	6.1 ± 0.6	< LOQ	9.8 ± 1.0
16	violaxanthin palmitate oleate	< LOQ	6.6 ± 0.9	10.9 ± 0.5	< LOQ	4.7 ± 0.1	< LOQ
17	antheraxanthin dilaurate	< LOQ	10.8 ± 0.2	5.4 ± 0.1	< LOQ	5.6 ± 0.6	< LOQ
18	antheraxanthin dimyristate	< LOQ	3.8 ± 0.2	0.9 ± 0.1	< LOQ	1.9 ± 0.1	< LOQ
19	luteoxanthin palmitate oleate	< LOQ	5.1 ± 0.3	1.2 ± 0.1	< LOQ	3.9 ± 0.1	< LOQ
20	zeinoxanthin palmitate oleate or β-cryptoxanthin palmitate oleate	< LOQ	1.1 ± 0.3	< LOQ	< LOQ	0.9 ± 0.1	< LOQ
21	violaxanthin palmitate stearate	< LOQ	16.1 ± 3.3	7.0 ± 1.1	< LOQ	10.1 ± 0.3	< LOQ
22	violaxanthin myristate	< LOQ	6.4 ± 3.3	2.8 ± 1.1	< LOQ	4.0 ± 0.3	< LOQ
**Total carotenoids (mg LE/100 g DW)**		**378** ± 17	**67** ± 4	**127** ± 9	**260** ± 4	**70** ± 4	**384** ± 4

**Table 3 plants-08-00384-t003:** Contents of lutein and β-carotene in different plant sources.

Plant Source	(all-*trans*)-β-carotene (mg/100 g DW)	(all-*trans*)-lutein (mg/100 g DW)	Ref.
Green leaves of Japanese knotweed	97	144	This study
Green leaves of Bohemian knotweed	69	97	This study
Spinach leaves	97	128	This study
Green leaves of Japanese knotweed	63	24	[35]
Spinach leaves	9–75	19–83	[43,44,45,46,47,48]
Kale leaves	30–40	36–55	[49,50]
Marigold petals	16	280	[51]
Carrot roots	11–56	10–31	[43,48]
White cabbage leaves	2–57	4–25	[43,45,48]
Broccoli crown	5–33	7–20	[45,48]
Cucumber fruits	5–17	1–24	[43,48]

**Table 4 plants-08-00384-t004:** Preparation of spiked leaf mixtures.

Sample	m [mg]	V_add_((all-*trans*)-β-carotene) [µL]^a^	V_add_((all-*trans*)-lutein) [µL]^a^	V_add_(acetone) [µL]
Spinach leaves	20	200	200	9600
Japanese knotweed green leaves	20	200	200	9600
Japanese knotweed green–yellowish leaves	30	100	30	9870
Japanese knotweed yellow leaves	30	100	50	9850
Bohemian knotweed green leaves	20	200	200	9600
Bohemian knotweed yellow leaves	30	100	30	9870

^a^ Concentration of (all-*trans*)-β-carotene and (all-*trans*)-lutein was 60 mg/L and 112 mg/L, respectively.

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
