# Peer review of "Japanese and Bohemian Knotweeds as Sustainable Sources of Carotenoids"

_plants, 2019, doi:10.3390/plants8100384_

Round 1

Reviewer 1 Report

There are only a few points where this paper could be improved:

Figure 1. Hopefully the photograph of silica gel plates will reproduce better in the journal. Orange color of β-carotene is not visible, most bands are uniform yellow-green.

Table 1. A “ _ “ sign is missing for recovery in yellow leaves of Bohemian knotweed. In case of recovery, only two values were obtained per determination, so the averages are presented with presumably absolute deviation? For content – are these values the averages and standard deviations of triplicate determinations? It should be mentioned in the title or legend under the table.

Table 3. The plant material for common vegetables should be qualified – marigold petals, carrot roots, cucumber fruits, etc. The ranges of values taken from references other than this study are extremely wide for spinach, carrots, cabbage, broccoli and cucumber. Are they truly representative for these vegetables, since very few references are quoted? Could it be due to a poor extraction method in some studies or the inferior variety (in respect to carotenoid content) of the vegetable in question?

Line 296. “Spiking” is incorrect for “the same volume of acetone” It was added or substituted ….

Line 391. It should be “harvesting”, not “excavation”. Presumably knotweed should be removed with roots by digging? The authors could develop the idea of “exploitation of plant parts (zero waste)”. After extraction, most of plant material remains – how they propose to utilize it? Feeding livestock? What about concentration of heavy metals?

References are not uniformly presented. A few examples – some titled are capitalized, some are not, Latin names are not always italicized and properly presented (line 522), some spaces are missing (line 527), names not properly abbreviated (line 517 – Shah H.U. and two more authors not mentioned, line 520 – Camp, J.V.). Careful editing is necessary.

Reviewer 2 Report

In this work, the authors analyze the carotenoid content of two species of invasive alien plants, and suggest that due to the high carotenoid content of the extracted biomasses, they can be used to obtain an economic benefit.

After reviewing the work, the only thing I can say is that it is impeccably done and explained. I am glad to tell the authors that I have no request to make. Congratulations.

Round 2

Reviewer 1 Report

The paper is now suitable for publication. The only required correction is that of "broccoli fruits" (Table 3). The edible part of broccoli is so-called "crown", i.e, flower buds, tightly bunched together (inflorescence), like in cauliflower. 

The explanation of using fresh weight (author's letter) is confusing. Hopefully the authors converted the results of quoted studies (if necessary) to dry weight (DW), since that is used in this study.

Author Response

The paper is now suitable for publication. The only required correction is that of "broccoli fruits" (Table 3). The edible part of broccoli is so-called "crown", i.e, flower buds, tightly bunched together (inflorescence), like in cauliflower.

The manuscript has been corrected according to the reviewer's comment.

The explanation of using fresh weight (author's letter) is confusing. Hopefully the authors converted the results of quoted studies (if necessary) to dry weight (DW), since that is used in this study.

Yes, the reviewer is correct. We apologise for the confusion. What we meant was, that for Table 3 we used the carotenoid content data reported for fresh (untreated) plant materials in order to make the comparisons of any value. All values are normalised to dry weight.

We would like to thank the reviewer again for all comments and corrections. We are sure that they further strengthen our manuscript.